# Apatite Formation on α-Tricalcium Phosphate Modified with Bioresponsive Ceramics in Simulated Body Fluid Containing Alkaline Phosphatase

**DOI:** 10.3390/biomimetics9080502

**Published:** 2024-08-20

**Authors:** Taishi Yokoi, Shinji Tomita, Jin Nakamura, Ayae Sugawara-Narutaki, Yuko Matsukawa, Masakazu Kawashita, Chikara Ohtsuki

**Affiliations:** 1Institute of Biomaterials and Bioengineering, Tokyo Medical and Dental University, 2-3-10 Kanda-Surugadai, Chiyoda-ku, Tokyo 101-0062, Japan; narutaki.ayae@tmd.ac.jp (A.S.-N.); kawashita.bcr@tmd.ac.jp (M.K.); 2Graduate School of Engineering, Nagoya University, Furo-cho, Chikusa-ku, Nagoya 464-8603, Japan; shinji056@gmail.com (S.T.); matsukawa.yuko.j0@f.mail.nagoya-u.ac.jp (Y.M.); ohtsuki@chembio.nagoya-u.ac.jp (C.O.); 3Graduate School of Life Science and Systems Engineering, Kyushu Institute of Technology, 2-4 Hibikino, Wakamatsu-ku, Kitakyushu-shi 808-0196, Japan; jin@life.kyutech.ac.jp

**Keywords:** phosphate ester, alkaline phosphatase, bioresponsive material, surface modification, tricalcium phosphate, apatite, simulated body fluid

## Abstract

Bioresponsive ceramics, a new concept in ceramic biomaterials, respond to biological molecules or environments, as exemplified by salts composed of calcium ions and phosphate esters (SCPEs). SCPEs have been shown to form apatite in simulated body fluid (SBF) containing alkaline phosphatase (ALP). Thus, surface modification with SCPEs is expected to improve the apatite-forming ability of a material. In this study, we modified the surface of α-tricalcium phosphate (α-TCP) using methyl, butyl, or dodecyl phosphate to form SCPEs and investigated their apatite formation in SBF and SBF containing ALP. Although apatite did not form on the surface of the unmodified α-TCP in SBF, apatite formation was observed following surface modification with methyl or butyl phosphate. When ALP was present in SBF, apatite formation was especially remarkable on α-TCP modified with butyl phosphate. These SCPEs accelerated apatite formation by releasing calcium ions through dissolution and supplying inorganic phosphate ions, with the latter process only occurring in SBF containing ALP. Notably, no apatite formation occurred on α-TCP modified with dodecyl phosphate, likely because of the low solubility of the resulting calcium dodecyl phosphate/calcium phosphate composites. This new method of using SCPEs is anticipated to contribute to the development of novel ceramic biomaterials.

## 1. Introduction

Ceramic biomaterials can be classified as bioinert, bioactive, and bioabsorbable ceramics [1]. Bioinert ceramics, such as alumina and zirconia, are used clinically as artificial joints [2] and tooth roots [3]. Bioactive ceramics, such as hydroxyapatite (HAp, stoichiometric composition: Ca_10_(PO_4_)_6_(OH)_2_) [4] and bioactive glasses [5], and bioabsorbable ceramics, such as α- and β-tricalcium phosphate (TCP; Ca_3_(PO_4_)_2_) [6] and octacalcium phosphate [7], have been used to repair bone defects. In contrast to these traditional classifications of ceramic biomaterials, we are proposing a new classification of ‘bioresponsive’ ceramics. We define bioresponsive ceramics as ceramics that respond to biological molecules and environments. We expect that this material concept will lead to the development of novel ceramic biomaterials. In addition to being a new research field in ceramic biomaterials, bioresponsive ceramics have the potential to open up new biomedical applications.

To establish the concept of bioresponsive ceramics, we have targeted bone-repair materials that respond to enzymes present in the human body. The use of various enzymes to synthesise ceramics has been reported, as exemplified by the synthesis of calcium carbonate [8,9,10] and calcium phosphate [11,12,13] using urease. Urease decomposes urea to carbon dioxide and ammonia. Calcium carbonate is produced by the reaction of calcium ions with the formed carbonate ions, and calcium phosphate precipitation is induced by the increase in the solution pH resulting from ammonia formation. In addition, calcium phosphate has been synthesised using alkaline phosphatase (ALP) [14,15,16,17,18,19]. ALP hydrolyses phosphate esters to form inorganic phosphate ions. ALP, which is found in human blood, is involved in the formation of bone minerals, namely, calcium phosphates, via matrix vesicle-mediated mineralisation [20]. Therefore, the artificial synthesis of calcium phosphates using ALP is an important biomimetic process, and research in this area will help advance the concept of bioresponsive materials.

We have investigated the reactions of salts composed of calcium ions and phosphate esters (SCPEs) in simulated body fluid (SBF) containing ALP. SBF is an aqueous solution used to estimate the bone-bonding properties of materials [21]. The guidelines for evaluating bone-bonding properties using SBF are stipulated by ISO 23317:2014 [22]. Materials that form apatite in SBF can potentially bond to bones after implantation in bony defects. Thus, the apatite-forming abilities of materials are commonly investigated using SBF [23,24,25,26,27]. We previously reported that calcium phenyl phosphate does not undergo significant reactions in SBF, whereas this SCPE forms apatite in SBF containing ALP [28]. In SBF, calcium phenyl phosphate gradually dissolves and releases calcium and phenyl phosphate ions. ALP decomposes the phenyl phosphate ions into inorganic phosphate ions and phenol. The subsequent reaction of the released calcium ions and formed phosphate ions results in apatite formation. In addition, the reaction rate for apatite formation from SCPEs can be varied by changing the alkyl group of the phosphate ester [29]. Owing to these ALP-responsive properties, SCPEs are bioresponsive materials with precisely controllable reaction rates.

Owing to these properties, the reactivity of a material will likely be improved upon combining with SCPEs. α-TCP sintered bodies have been reported to exhibit negligible dissolution in SBF. Furthermore, apatite does not form on α-TCP sintered bodies in SBF [30]. The rate-limiting step in the transformation of α-TCP into apatite is likely apatite nucleation. Therefore, we expect that the reactivity of α-TCP sintered bodies can be improved via surface modification with bioresponsive ceramics that undergo apatite nucleation. In this study, this hypothesis was tested by treating α-TCP sintered bodies with three types of phosphate esters to form bioresponsive ceramics on the sintered body surface. Furthermore, the reaction characteristics in SBF and SBF containing ALP were compared.

## 2. Materials and Methods

### 2.1. Preparation of α-TCP Sintered Bodies and Surface Modification with Phosphate Esters

Calcium carbonate (CaCO_3_; Nacalai Tesque, Inc., Kyoto, Japan) and calcium hydrogen phosphate dihydrate (CaHPO_4_·2H_2_O; Nacalai Tesque, Inc.) powders were mixed at a Ca/P molar ratio of 3:2. The mixed powder was heat treated at 1400 °C for 12 h using an electric furnace (Super-Burn, Motoyama Co., Ltd., Osaka, Japan) and then cooled to room temperature. The heat-treated powder was crushed to obtain α-TCP powder.

The α-TCP powder (500 mg) was uniaxially pressed at 100 MPa for 2 min using a mould with a diameter of 13 mm. Compacted α-TCP was sintered in the electric furnace using the following heat-treatment process: (1) increase the temperature to 1400 °C; (2) decrease the temperature to 1300 °C and maintain the sample at this temperature for 12 h; (3) cool to room temperature. A heating rate of 5 °C/min was used in this synthetic process.

Methyl phosphate (mono- and diester mixture; Tokyo Chemical Industry Co., Ltd., Tokyo, Japan), butyl phosphate (mono- and diester mixture; Tokyo Chemical Industry Co., Ltd.), or dodecyl phosphoric acid (FUJIFILM Wako Pure Chemical Co., Osaka, Japan) was dissolved in ethanol to prepare a 0.5 mol/dm^3^ (M) phosphate ester solution. In the case of methyl phosphate, which had a monoester content of 50.0–55.0%, the solution was prepared assuming a monoester content of 50.0%. In the case of butyl phosphate, which had a monoester content of 34.0–45.0%, the solution was prepared assuming a monoester content of 34.0%. After placing 5 cm^3^ of the prepared phosphate ester solution in a glass bottle, four α-TCP sintered bodies were immersed in the solution while leaning against the bottle wall to allow for the reaction of the entire sintered body surface. After reacting at 36.5 °C for 5 d, the samples were washed with ethanol and dried at 40 °C. The obtained phosphate ester-treated α-TCP sintered bodies were named according to the phosphate ester used and treatment period, as shown in Table 1.

### 2.2. Preparation of SBF and Sample Soaking

SBF was prepared according to our previously reported method [28] using sodium chloride (NaCl), sodium hydrogen carbonate (NaHCO_3_), potassium chloride (KCl), dipotassium hydrogen phosphate (K_2_HPO_4_·3H_2_O), magnesium chloride (MgCl_2_·6H_2_O), calcium chloride (CaCl_2_), and sodium sulphate (Na_2_SO_4_), all of which were obtained from Nacalai Tesque Inc. The pH of the SBF was adjusted to 7.4 at 36.5 °C using solid tris(hydroxymethyl)aminomethane (NH_2_C(CH_2_OH)_3_; Nacalai Tesque Inc.) and a 1.0 M solution of hydrochloric acid (HCl). The HCl solution was prepared by diluting 35 mass% hydrochloric acid (Wako Pure Chemical Industries, Ltd., Osaka, Japan).

The unmodified α-TCP sintered bodies and α-TCP sintered bodies treated with phosphate esters were soaked in 25 cm^3^ of SBF or SBF containing ALP at 36.5 °C for 1, 3, and 7 d. To verify the effects of ALP, 100 mm^3^ of an aqueous solution containing 5.0 units of ALP (from bovine intestinal mucosa, Sigma-Aldrich Japan, Co., Tokyo, Japan) was added to the SBF daily. After soaking for 1, 3, and 7 d, the samples were removed from the SBF, washed with ultrapure water, and then dried at 40 °C.

### 2.3. Characterisation

The crystalline phases of the samples before and after soaking in the SBF were investigated using X-ray diffraction (XRD; RINT-2100/PC, Rigaku Co., Tokyo, Japan). The XRD patterns were collected via the 2*θ*/*θ* method using CuKα radiation. The obtained XRD patterns were analysed using PDXL2 software ver. 2.9.1.0 (Rigaku). The surface morphologies of the samples before and after soaking in SBF were observed using scanning electron microscopy (SEM; JSM-5600, JEOL Ltd., Tokyo, Japan). Before observation, the samples were coated with a thin gold film using a fine coater (JFC-1200, JEOL Ltd.) at an internal pressure of 8 Pa and a current of 20 mA for 200 s. These deposition conditions produced a gold film thickness of approximately 30 nm. The Ca and P concentrations in the SBF were determined using inductively coupled plasma atomic emission spectroscopy (ICP-AES; Optima 2000 DV, PerkinElmer Japan, Kanagawa, Japan).

## 3. Results and Discussion

### 3.1. α-TCP Sintered Bodies without Phosphate Ester Modification

The crystalline phases of the α-TCP sintered bodies before and after soaking in SBF or SBF containing ALP for 7 d were characterised using XRD. The XRD patterns of the samples are shown in Figure 1. Before soaking in SBF, the α-TCP sintered body exhibited a single crystalline phase corresponding to α-TCP. The powder diffraction file (PDF) #00-009-0348 was used to identify the α-TCP. Furthermore, no significant differences were observed between the XRD pattern of the α-TCP sintered body before soaking and those of the α-TCP sintered bodies soaked in the SBF or SBF containing ALP for 7 d.

The surface morphologies of the unmodified α-TCP sintered bodies before and after soaking in the SBF or SBF containing ALP for 7 d are shown in Figure 2. The sintering of the α-TCP particles with sizes of several micrometres was observed in the sample before soaking. In addition, this sample contained pores with sizes of approximately 1 μm. The porosity of the sample, as calculated from the bulk and true densities, was 19%. The structures of the samples soaked in the SBF or SBF containing ALP for 7 d were similar to that of the α-TCP sintered body before soaking. Furthermore, no new precipitates were observed after soaking these samples.

The XRD and SEM results (Figure 1 and Figure 2) indicate that the unmodified α-TCP sintered body is stable in SBF and SBF containing ALP. As discussed below, the changes in the Ca and P concentrations in the SBF and SBF containing ALP suggest that the unmodified α-TCP sintered body was poorly soluble in these solutions and did not induce calcium phosphate formation. Hence, the unmodified α-TCP sintered body does not exhibit apatite formation in SBF, regardless of the presence of ALP.

### 3.2. α-TCP Sintered Bodies Modified with Methyl Phosphate (MeP5d)

The crystalline phases of the MeP5d samples before and after soaking in the SBF or SBF containing ALP for various periods were characterised using XRD (Figure 3). For the MeP5d before soaking, the only crystalline phase detected was α-TCP. Even when this sample was soaked in SBF, only α-TCP was observed until 3 d. After 7 d, reflection peaks assigned to HAp were detected in addition to those assigned to α-TCP. The PDF #00-009-0348 was used to identify the HAp. For the MeP5d soaked in SBF containing ALP, weak reflection peaks attributed to HAp were detected after 1 d as well as those corresponding to α-TCP. The intensities of the HAp reflection peaks increased when the soaking period was prolonged.

The surface morphologies of the MeP5d samples before and after soaking in SBF or SBF containing ALP for various periods are shown in Figure 4. Before soaking, precipitates with sizes of 1–1.5 μm were observed on the surface of the MeP5d. Soaking in SBF for 1 d did not cause any significant changes in the morphology of these precipitates. In contrast, after 3 and 7 d soaking in SBF, the MeP5d surface was covered with precipitates. In the SBF containing ALP, the MeP5d exhibited higher reactivity. Consequently, the surface of the MeP5d was covered with precipitates after 1 d, but the surface morphology showed no further significant changes after soaking in the SBF containing ALP for 3 and 7 d.

Based on the XRD and SEM results in Figure 3 and Figure 4, the newly formed precipitates on the surface of the MeP5d soaked in SBF for 7 d or soaked in SBF containing ALP for 1, 3, and 7 d were apatite. In addition, the observed apatite had the typical hemispherical morphology of apatite particles formed in SBF [1]. Notably, the methyl phosphate treatment enhanced the apatite formation on the α-TCP sintered bodies. Although the precipitates observed on the MeP5d surface before soaking could not be identified using XRD, they are likely calcium methyl phosphate (CaMeP). CaMeP can dissolve in SBF, thereby releasing calcium ions and increasing the degree of supersaturation with respect to HAp. Consequently, apatite formation is accelerated. The XRD patterns (Figure 3) and SEM images (Figure 4) indicate that more apatite was formed in the SBF containing ALP than in SBF alone. This finding is reasonable because apatite nucleation and crystal growth can be accelerated by not only the release of calcium ions from CaMeP via dissolution but also the hydrolysis of methyl phosphate by ALP, which increases the concentration of inorganic phosphate ions in SBF.

### 3.3. α-TCP Sintered Bodies Modified with Butyl Phosphate (BuP5d)

Figure 5 shows the XRD patterns of the BuP5d samples before and after soaking in SBF or SBF containing ALP for various periods. The XRD pattern of the BuP5d before soaking in SBF revealed the presence of calcium butyl phosphate (CaBuP) and dicalcium phosphate anhydrous (DCPA; CaHPO_4_) in addition to α-TCP. The PDF #00-009-0080 was used to identify the DCPA. The CaBuP phase was identified based on previously reported data [29]. Upon soaking the BuP5d in SBF, the DCPA phase disappeared after 1 d and reflection peaks derived from HAp were detected. The intensities of the HAp reflection peaks increased when the soaking period was prolonged. For the BuP5d in SBF containing ALP, the DCPA phase disappeared after soaking for 1 d, and reflection peaks attributed to HAp appeared after soaking for 3 and 7 d. Similar to the trend observed for the sample soaked in SBF, the intensities of the HAp reflection peaks increased upon increasing the soaking period in SBF containing ALP.

Figure 6 shows the surface morphologies of the BuP5d before and after soaking in SBF or SBF containing ALP for various periods. The surface of the BuP5d before soaking appeared to be covered with a film-like material instead of the precipitates observed on the MeP5d surface. After soaking for 1 and 3 d in the SBF, the sample surface was covered with hemispherical precipitates. The size of these precipitates increased when the BuP5d sample was soaked in the SBF for 7 d. For the BuP5d soaked in SBF containing ALP, the sample surface was completely covered with precipitates after 1 d and the size of the precipitates increased when the soaking period was longer.

Based on the XRD patterns (Figure 5) and SEM images (Figure 6), the newly formed hemispherical precipitates on the surfaces of the soaked samples were apatite. Similar to the methyl phosphate treatment, the butyl phosphate treatment accelerated apatite formation on the α-TCP sintered bodies. The mechanism responsible for accelerated apatite formation in this case is likely the same as that proposed for the methyl phosphate treatment. Interestingly, based on the XRD analysis, butyl phosphate has a greater accelerating effect on apatite formation than methyl phosphate. For the MeP5d soaked in SBF containing ALP for 7 d, the HAp reflection peak intensity at approximately 33° is almost the same as the α-TCP reflection peak intensity at approximately 31° (Figure 3b). In contrast, as shown in Figure 5b, for the BuP5d soaked in SBF containing ALP for 7 d, the HAp reflection peak at approximately 33° had a higher intensity than the α-TCP reflection peak at approximately 31°. In the case of the methyl phosphate treatment, the surface of the α-TCP sintered body was only partially covered in precipitates (Figure 4, before soaking), whereas, in the case of the butyl phosphate treatment, the α-TCP sintered body was completely enveloped (Figure 6, before soaking). In other words, butyl phosphate was present on the entire surface of the α-TCP sintered body, largely in the form of CaBuP, which likely promoted apatite formation.

### 3.4. α-TCP Sintered Bodies Modified with Dodecyl Phosphate (DoP5d)

XRD was used to characterise the crystalline phases of the DoP5d samples before and after soaking in SBF or SBF containing ALP for various periods (Figure 7). Before soaking in SBF, the DoP5d exhibited reflection peaks derived from calcium dodecyl phosphate/calcium phosphate composites (CaDoP/CaP), as well as other unidentified peaks. The CaDoP/CaP phase was identified based on a previous report [31]. Similar reflection peaks were observed after soaking the DoP5d in SBF for 1, 3, and 7 d. In addition, no significant changes in the reflection peaks occurred when the DoP5d was soaked in SBF containing ALP for 1, 3, and 7 d. Notably, none of the DoP5d samples exhibited reflection peaks derived from α-TCP.

The SEM images in Figure 8 show the surface morphologies of the DoP5d before and after soaking in SBF or SBF containing ALP for various periods. The surface of the DoP5d sample before soaking appeared to be covered with plate-shaped precipitates with lengths of 5–10 μm and widths of approximately 1 μm. No significant morphological changes occurred after soaking in the SBF for up to 7 d. In addition, unlike the MeP5d and BuP5d (Figure 4b and Figure 6b), no hemispherical apatite crystals were observed for the DoP5d. In the SBF containing ALP, the plate-shaped precipitates observed on the DoP5d before soaking became fibrous as the soaking period was prolonged, and no hemispherical apatite crystals were observed.

The XRD and SEM analyses (Figure 7 and Figure 8) demonstrate that the DoP5d had lower reactivity than the MeP5d and BuP5d, which was consistent with the results of a previous study on the behaviour of CaDoP in SBF and SBF containing ALP [29]. Because of its low solubility, CaDoP likely suppressed the reactivity of the α-TCP sintered body in SBF.

### 3.5. Sample Reactivity Based on Changes in the Ca and P Concentrations in SBF and SBF Containing ALP

Changes in the Ca and P concentrations can provide useful insights into sample behaviour in SBF or SBF containing ALP. Figure 9 shows the time-dependent changes in the Ca and P concentrations of the SBF and SBF containing ALP during the soaking of the unmodified and phosphate ester-modified α-TCP sintered bodies. For all of the samples, the Ca concentration is regarded as the Ca ion concentration in the solution. For the unmodified α-TCP sintered body, the P concentration is equivalent to the concentration of orthophosphate ions in the solution. In contrast, as the MeP5d, BuP5d, and DoP5d samples contain phosphate esters that can be released into the solution, the P concentration is equivalent to the sum of the orthophosphate ion and phosphate ester concentrations in the solution. Here, we first discuss MeP5d and BuP5d, for which apatite formation was observed, and then the unmodified α-TCP sintered body and DoP5d, which did not exhibit apatite formation.

In the case of the MeP5d in SBF, the Ca concentration gradually decreased, reaching approximately 2.1 mM after 3 d and then remaining constant. In contrast, the P concentration increased significantly from 1 mM to 2.6 mM before remaining nearly constant. For the MeP5d in SBF containing ALP, the Ca concentration gradually decreased to 1.3 mM after 3 d and then remained constant. However, the P concentration initially exhibited a dramatic increase to 2.8 mM before gradually decreasing, reaching 2.6 mM after 7 d.

For the BuP5d in SBF, the Ca concentration gradually decreased to approximately 2.1 mM after 3 d and then became constant. However, the P concentration showed an initial significant increase to 2.6 mM before remaining almost constant. In the case of the BuP5d in SBF containing ALP, the Ca concentration gradually decreased and reached 1.4 mM after 7 d. Conversely, the P concentration first increased to 2.2 mM and then gradually decreased to 1.7 mM after 7 d.

Similar time-dependent changes in the Ca and P concentrations were observed for the MeP5d and BuP5d in SBF and SBF containing ALP. The Ca and P concentrations of both samples reached steady states after 3 d. However, during this time, apatite formation occurred on these samples, as revealed by the XRD results (Figure 3 and Figure 5) and SEM observations (Figure 4 and Figure 6). These results imply that the decreases in the Ca and P concentrations resulting from apatite formation were balanced by increases in the Ca and P concentrations owing to the dissolution of the SCPEs.

The Ca concentration in the SBF containing ALP was slightly lower for the MeP5d than for the BuP5d after 1 and 3 d. The decrease in the Ca concentration was attributed to the formation of calcium phosphate; thus, the MeP5d exhibited a faster calcium phosphate formation rate than the BuP5d. Such a result seems contradictory to the XRD results (Figure 3b and Figure 5b); the HAp reflection peak of the BuP5d was clearly more intense than that of the MeP5d after 1 d, indicating that the apatite formed on the MeP5d at 1 d had a lower crystallinity than that formed on the BuP5d. The P concentration of the MeP5d was higher than that of the BuP5d; however, the P concentration is the sum of the concentrations of inorganic phosphate ions and phosphate esters. Hence, this difference is likely due to the solubility of CaMeP being higher than that of CaBuP [29].

As shown in Figure 9, the Ca and P concentration changes observed with the unmodified α-TCP sintered body and DoP5d were similar. After soaking for 1 d in SBF or SBF containing ALP, the Ca concentrations decreased from 2.5 mM to approximately 2 mM, and then remained almost constant. No changes in the P concentrations were observed during sample soaking for up to 7 d.

The decrease in the Ca concentrations of these two samples likely results from the adsorption of calcium ions on the sample surfaces in SBF and SBF containing ALP. The observed Ca and P concentration changes indicate that the dissolution of the unmodified α-TCP sintered body and DoP5d samples was negligible, and that these samples did not react significantly in SBF or SBF containing ALP, indicating that no apatite formation occurred, consistent with the XRD and SEM results (Figure 1, Figure 2, Figure 7 and Figure 8).

## 4. Conclusions

Ceramic biomaterials are traditionally classified as bioinert, bioactive, and bioabsorbable ceramics. In contrast to these classifications, we are proposing bioresponsive ceramics as a new material concept, which we expect to lead to the development of novel ceramic biomaterials. In particular, combining bioresponsive materials with other materials can lead to improvements in biological properties. In this study, we investigated the reactivity of α-TCP sintered bodies modified with bioresponsive ceramics, namely, SCPEs, in SBF and SBF containing ALP. The unmodified α-TCP sintered body could not form apatite on its surface in SBF. However, apatite was formed on the α-TCP sintered bodies modified with methyl and butyl phosphates in SBF. Moreover, apatite formation on the α-TCP sintered bodies modified with methyl and butyl phosphates was accelerated when ALP was added to SBF. These findings are reasonable, as apatite nucleation and crystal growth can be accelerated by calcium ions released during the dissolution of these SCPEs and also by hydrolysis of the constituent phosphate esters by ALP, which would increase the inorganic phosphate ion concentration in SBF. In particular, remarkable apatite formation was observed on the α-TCP sintered body modified with butyl phosphate in SBF containing ALP. Therefore, surface modification using methyl and butyl phosphates imparted apatite-forming abilities to α-TCP; however, based on the current data, whether α-TCP transformation to HAp occurred or not is still unclear. This point requires further investigation and should be included in future studies. In contrast, no apatite formation occurred on the α-TCP sintered body modified with dodecyl phosphate in SBF or SBF containing ALP. These findings are meaningful for establishing a methodology that can impart α-TCP sintered bodies with apatite-forming characteristics using bioresponsive ceramics or improve the inherent apatite-forming ability. This surface modification method is also expected to be applicable to a variety of other ceramic materials as well as polymeric materials and contribute to advancing the field of ceramic biomaterials.

## Figures and Tables

**Figure 1 biomimetics-09-00502-f001:**
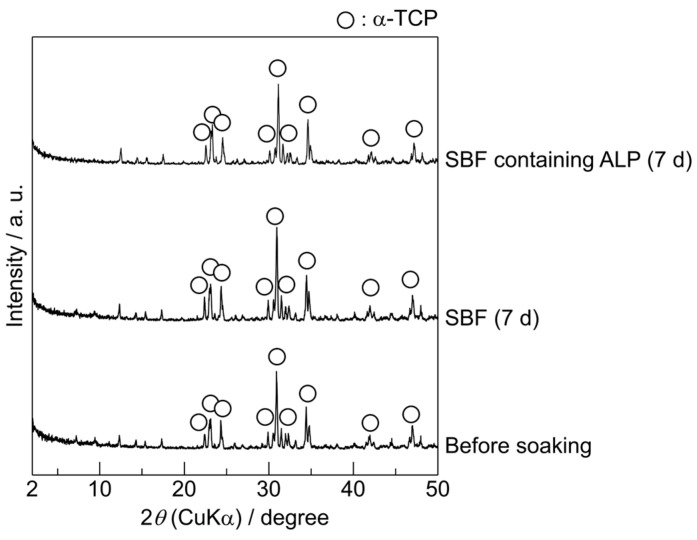
X-ray diffraction (XRD) patterns of α-tricalcium phosphate (α-TCP) sintered bodies before and after soaking in the simulated body fluid (SBF) and SBF containing alkaline phosphatase (ALP) for 7 d.

**Figure 2 biomimetics-09-00502-f002:**
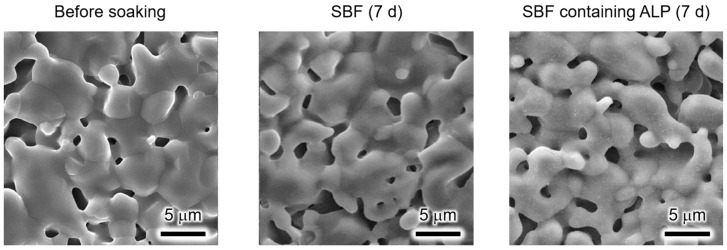
Scanning electron microscopy (SEM) images of α-TCP sintered bodies before and after soaking in the SBF or SBF containing ALP for 7 d. (Observation magnification: ×3000).

**Figure 3 biomimetics-09-00502-f003:**
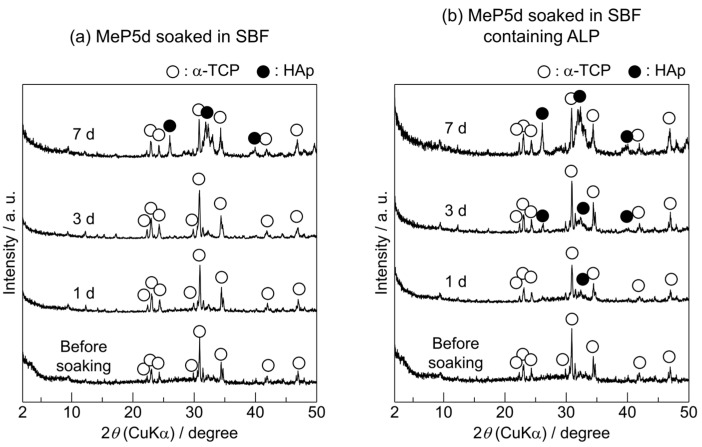
XRD patterns of the α-TCP sintered bodies modified with methyl phosphate (MeP5d) before and after soaking in (**a**) SBF and (**b**) SBF containing ALP for various periods.

**Figure 4 biomimetics-09-00502-f004:**
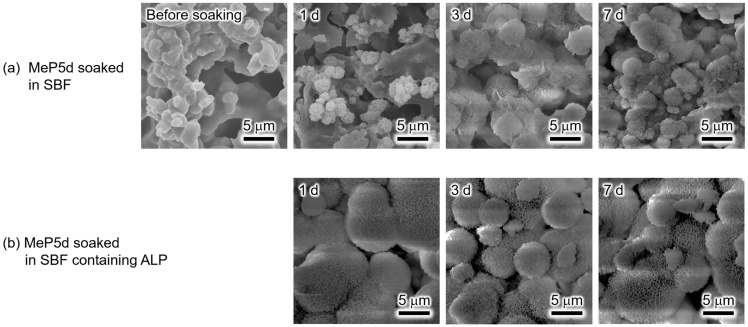
SEM images of the MeP5d before and after soaking in (**a**) SBF and (**b**) SBF containing ALP for various periods. (Observation magnification: ×3000).

**Figure 5 biomimetics-09-00502-f005:**
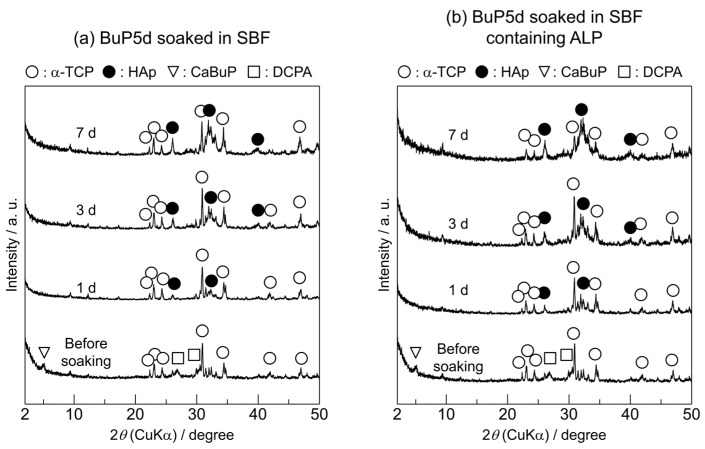
XRD patterns of the α-TCP sintered bodies modified with butyl phosphate (BuP5d) before and after soaking in (**a**) SBF and (**b**) SBF containing ALP for various periods.

**Figure 6 biomimetics-09-00502-f006:**
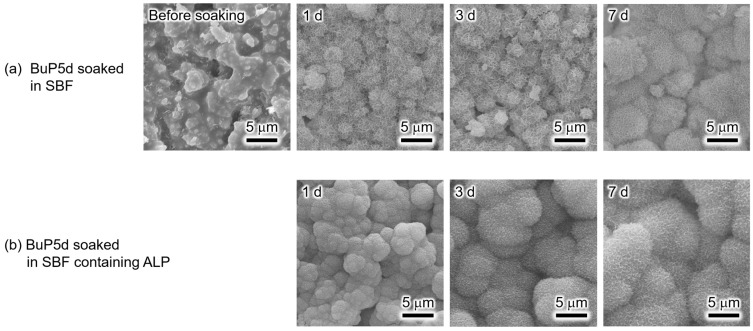
SEM images of the BuP5d before and after soaking in (**a**) SBF and (**b**) SBF containing ALP for various periods. (Observation magnification: ×3000).

**Figure 7 biomimetics-09-00502-f007:**
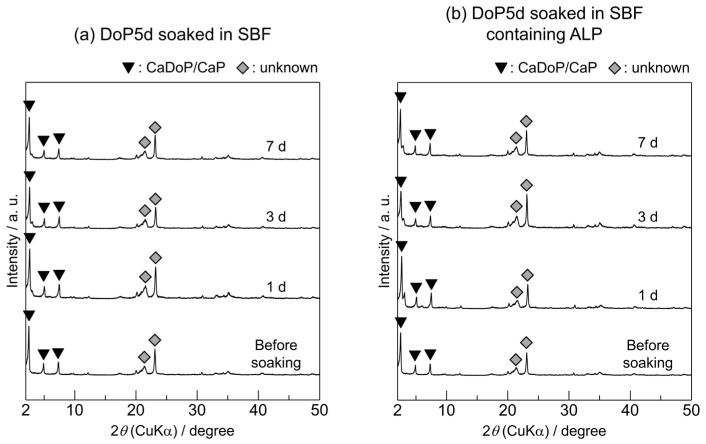
XRD patterns of the α-TCP sintered bodies modified with dodecyl phosphate (DoP5d) before and after soaking in (**a**) SBF and (**b**) SBF containing ALP for various periods.

**Figure 8 biomimetics-09-00502-f008:**
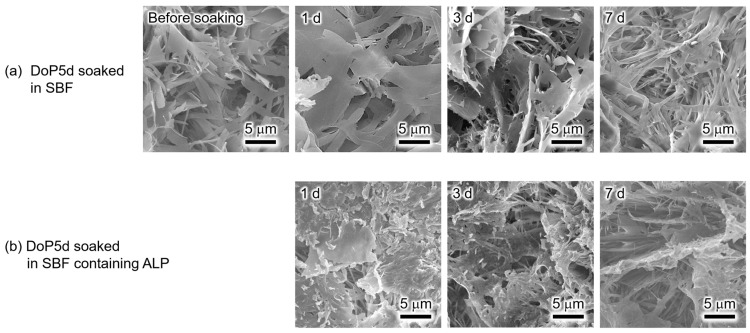
SEM images of the DoP5d before and after soaking in (**a**) SBF and (**b**) SBF containing ALP for various periods. (Observation magnification: ×3000).

**Figure 9 biomimetics-09-00502-f009:**
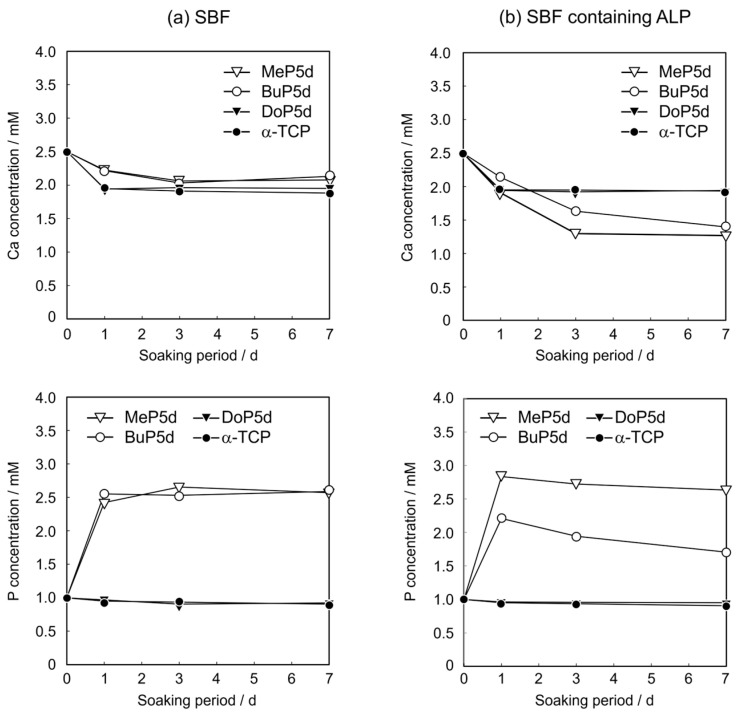
Time-dependent Ca and P concentrations in (**a**) SBF and (**b**) SBF containing ALP during the soaking of unmodified and phosphate ester-modified α-TCP sintered bodies.

**Table 1 biomimetics-09-00502-t001:** Sample name definitions.

Sample Name	Phosphate Ester	Treatment Period (d)
MeP5d	Methyl phosphate	5
BuP5d	Butyl phosphate	5
DoP5d	Dodecyl phosphate	5

## Data Availability

The original contributions presented in the study are included in the article; further inquiries can be directed to the corresponding author.

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
