# Peer review of "Apatite Formation on α-Tricalcium Phosphate Modified with Bioresponsive Ceramics in Simulated Body Fluid Containing Alkaline Phosphatase"

_biomimetics, 2024, doi:10.3390/biomimetics9080502_

Round 1
Reviewer 1 Report
Comments and Suggestions for Authors
This manuscript is clearly presented. There are several technical comments on experimental results.
1. Please provide the crystal planes of XRD peaks in Fig. 1, 3, 5, and 7 for all known chemicals. The PDF# for corresponding crystal planes and software (company name and version) for fitting should also be provided. I suggest having tables of peak information for each figure. This is important for the credential of experimental data.
2. Please specify the magnification of all SEM images. I think the images are quite good even shrunk to such small-size photos.
3. Is there any usual process (absorption) or “melting” of α-TCP in Fig. 2? It seems that those particles were “melted” together even before soaking. Please re-examine these images and provide some explanations.
4. In Fig. 9, cases of unmodified and phosphate ester-modified α-TCP have almost constant concentrations of Ca and P implying the stability of both materials in SBF whether with ALP or not. However, I think the increased concentration of P may need more investigation, at least from a qualitative perspective.
Please note that methyl and butyl phosphates treated with α-TCP could release calcium and phosphate ions, but I speculate that the amount of release may not be high.
It may be possible to have a short discussion on the release of calcium phosphates based on the reaction's kinetic or thermodynamic view. I understand this kind of discussion is only at the preliminary level as the immersion test is already complicated in the chemical compassions of solution.
5. It may be interesting to see some higher magnification (smaller scales) of the SEM photos of the 7-day results for (1) MeP5d, (2) BuP5d, and (3) DoP5d samples soaked in SBF containing ALP. There should be some interesting morphologies of various chemicals on the surface of α-TCP.
Comments on the Quality of English LanguageIt may need a language check on typos or usage before final approval for publication.
Author Response
Please refer to the attached response letter.

Reviewer 2 Report
Comments and Suggestions for Authors
Comments and suggestions for the authors are enclosed in a PDF file.

Author Response

(The authors gave the same response as above.)
